# Lanthanum Carbonate Opacities—A Systematic Review

**DOI:** 10.3390/diagnostics12020464

**Published:** 2022-02-11

**Authors:** Jan Kampmann, Nina Pirschel Hansen, Anders Nikolai Ørsted Schultz, Andreas Hjelm Brandt, Frans Brandt

**Affiliations:** 1Medical Research Unit, Department of Internal Medicine, Hospital of Southern Jutland, 6200 Aabenraa, Denmark; ANOS@rsyd.dk (A.N.Ø.S.); FBK@rsyd.dk (F.B.); 2Institute of Regional Health Research (IRS), University of Southern Denmark, 5230 Odense, Denmark; 3Department of Radiology, Hospital of Southern Jutland, Aabenraa Branch, 6200 Aabenraa, Denmark; Nina.Pirschel.Hansen@rsyd.dk; 4Department of Radiology, Rigshospitalet, 2100 Copenhagen, Denmark; andreaskr5@gmail.com

**Keywords:** lanthanum carbonate, image studies, CKD

## Abstract

Background: Lanthanum carbonate is a phosphate binder used in advanced kidney disease. Its radiopaque appearance has been described in many case studies and case series. Misinterpretation of this phenomenon leads to unnecessary diagnostic tests and procedures. The objectives of this study were to summarize the literature on lanthanum carbonate opacities and present a visual overview. Methods: A systematic search was conducted using MEDLINE, Embase, and Web of Science. We included all types of studies, including case reports/studies, describing radiological findings of lanthanum carbonate opacities in patients with chronic kidney disease. No filter for time was set. Results: A total of 36 articles were eligible for data extraction, and 33 articles were included in the narrative synthesis. Lanthanum carbonate opacities were most commonly reported in the intestines (26 studies, 73%), stomach (8 studies, 21%), and the aerodigestive tract (2 studies, 6%). The opacities in the intestine were most frequently described as multiple, scattered radiopaque densities, compared with the aerodigestive tract, where the opacities were described as a single, round foreign body. Suspicion of contrast medium or foreign bodies was the most common differential diagnosis. LC opacities in patients with CKD are commonly misinterpreted as foreign bodies or suspect contrast media. Conclusions: CKD patients treated with LC may have opacities throughout the digestive tract that can vary in appearance. Stopping LC treatment or changing to an alternative phosphate binder prior to planned image studies can avoid diagnostic confusion. If this is not an option, knowledge of the presentation of LC opacities is important.

## 1. Introduction

Phosphate retention exacerbates renal osteodystrophy and increases the risk of soft tissue and vascular calcification and death [1]. Therefore, medical treatment includes phosphate binders [2]. Phosphate binders are used to control hyperphosphatemia [3]. They can be divided into calcium-containing phosphate binders such as calcium carbonate and calcium acetate. In cases of increased calcium levels, calcium-free phosphate binders, including the most widely used calcium-free phosphate binders sevelamer and lanthanum carbonate (LC), are used.

LC Lanthanum carbonate was introduced in 2005 in the United States, and in 2009 in Japan, under the brand name Fosrenol™ [4,5]. LC is available as a chewable tablet or oral powder [6]. It disassociates in the upper gastrointestinal tract to lanthanum ions (La^3+^), forming insoluble lanthanum phosphate complexes that pass through the GI tract almost unabsorbed [6,7]. The absorbed part is excreted in bile [7]. The appearance of LC-associated opacities’ was first described in 2006, suggesting that the opacities were related to calcium-phosphate accumulation [8]. However, it has since become clear that LC is responsible for the opacities [9]. The atomic number in LC is relatively high, at 57, and therewith close to barium at 56, which could explain the similarity to contrast signals. Other studies have described how the opacities in various anatomical regions with different image procedures can cause confusion with differential diagnosis. In worst cases, misinterpretation leads to unnecessary follow-up, delayed operations or procedures, and even surgical interventions [10,11,12,13].

This study was designed to identify the most common differential diagnosis and pitfalls when LC opacities are misinterpreted. In addition, we provide a visual overview of the varying appearances of LC in the digestive tract. Our systematic review is the first to assess LC opacities’ locations and appearances on different image modalities. Finally, we discuss directions for future research.

## 2. Materials and Methods

A systematic review was performed in accordance with the PRISMA statement [14]. The review was registered in advance at Prospero with the registration number CRD42020204971.

### 2.1. Eligibility

We included peer-reviewed studies of the radiological findings of chronic kidney disease patients treated with lanthanum carbonate opacities and published in English. We excluded posters, reviews or abstracts, or papers published in a language other than English.

### 2.2. Search Strategy and Study Selection

A database search in MEDLINE, Embase, and Web of Science was conducted on 23 September 2020. The search consisted of two blocks—one for chronic kidney disease (CKD) and one for LC. The two blocks were combined with the Boolean operator *AND* and modified accordingly for each database. The full search strategy can be found in Appendix A. There were no filters used in relation to time. The OVID™ filter was used to locate articles published in English to identify human studies only.

Articles were uploaded to Covidence™, and duplicates were removed. All studies were screened by title and abstract and, if eligible, for full text, by two independent investigators (J.K. and N.P.H.). A.N.Ø.S. was consulted as a referee in case of disagreement.

### 2.3. Data Extraction and Analysis

The following data regarding study characteristics were included in the extraction tables: publication year, country, study type, number of patients and image modality, opacity characteristics, location of opacity and differential diagnosis, type of exposure, and duration. The data were extracted using an Excel prespecified data abstraction form and piloted and evaluated after the first couple of studies had been extracted. Data extraction was performed independently (J.K. and N.P.H.), and in case of disagreement, A.N.Ø.S. was consulted.

Data analysis was conducted using narrative synthesis according to the guidelines for Synthesis without Meta-analysis in Systematic Reviews [15]. Studies were grouped according to exposure time, type of image modality and location of the opacity, and the characteristics of the opacity. Results of the analysis are described in absolute numbers or in percentages.

### 2.4. Quality Assessment

The quality of the studies was assessed by J.K. and N.P.H. using the “Methodological quality and synthesis of case series and case reports” chart [16]. The chart validated quality using eight questions.

Publications that did not ascertain the outcome (question 2), did not rule out alternative causes (question 4), and did not describe the case in full detail (question 8) were excluded.

The use of the chart for quality assessment of the cohort studies was discussed by J.K. and N.P.H. and found suitable. The aim of this study was to assess the most common differential diagnosis and pitfalls with LC opacities, for which the quality of the studies would be sufficient if they fulfilled the criteria listed in the chart. Hence, the chart was used for all included studies.

## 3. Results

A database search identified 1412 studies after the removal of duplicates. The full-text screening was conducted for 80 studies, of which 36 were included for data extraction. After quality extraction, 3 papers were excluded, resulting in 33 studies for the narrative synthesis (see Figure 1).

The narrative synthesis consisted of 30 [8,9,10,11,12,13,17,18,19,20,21,22,23,24,25,26,27,28,29,30,31,32,33,34,35,36,37,38,39,40] single case reports, 1 case series [4], and 2 cohort studies [41,42]. The case series included 9 patients, and the cohort studies included 126 and 169 patients.

Therefore, the review consists of reports from a total of 334 patients. The studies were from 14 different countries, with Japan being the most common [4,24,28,31,32,41].

Among the included papers, 13 studies reported dispensing LC in tablet form, but the remaining articles did not specify the dispensation form. The duration of LC treatment ranged from days [12] to 6 years [31]; 19 articles did not specify duration treatment.

Characteristics of included studies can be found in Table 1.

### 3.1. Quality of Included Studies

All but three articles passed the quality assessment. Both of the cohort studies were included and the case series article as well. The quality assessment is displayed in Table 2.

### 3.2. Type of Image Modality

The most frequent image modality was X-ray, reported in 73% of the papers [8,9,10,11,13,17,18,19,20,21,22,23,25,26,27,29,33,34,35,36,37,38], and CT, reported in 49% of the papers [4,8,9,11,18,19,21,24,26,28,31,32,33,37,41]. Two or more types of radiological imaging were reported in 33% of the papers [8,9,11,18,19,21,26,28,29,33,37].

Location, characteristics of opacity, and differential diagnosis

LC opacities were most commonly reported in the intestines (26 studies, 73%) [4,8,10,11,12,13,17,18,19,21,23,24,25,26,27,28,29,30,33,34,35,36,38,39,40,41,42], with 24% of all of the studied papers specifying the colon [13,23,25,30,34,36,37,39]. LCs were also reported in the stomach, by 21% of all studies [9,19,21,31,32,40,41], or the aerodigestive tract, by 6% of the papers [20,22].

The most frequently reported characteristics of LC in the intestines were *multiple*, *scattered radiopaque densities* [4,8,10,11,13,17,18,21,23,25,26,27,29,30,33,34,35,36,37,38,39,40,42], compared with LC opacities in the aerodigestive tract, where LC was reported as a *single*, *round foreign body*, with one of the papers describing it as *coin shaped* [20,22]. Three publications described the LC opacities as *multiple*, *hyperdense elements with sharp edges and beam hardening* [9,12,24]. In addition, there was a single case whereby the whole tablet was seen in the terminal ileum together with digested residues in the colon diverticulum [4], and another case in which the opacity is described as a radiopaque puddle with dilation and fecal impaction [19]. All LC opacities from the colon were described as *multiple*, *scattered radiopaque densities* [13,23,25,30,34,36,37,39].

Nine (38%) of the papers reporting LC opacities in the intestines also reported suspicion for contrast medium as a differential diagnosis [25,27,30,34,35,36,37,38,39]. All of these cases described *multiple*, *scattered radiopaque densities*. For detailed descriptions of varying radiological findings of LC opacities in the stomach [9,19,21,31,32,40,41], see Table 3, and for a visual overview, see Figure 2, Figure 3 and Figure 4.

Beam-hardening artifacts on computed tomography images are shown, which were caused by lanthanum carbonate hydrate in a patient on dialysis. Reprinted with permission from ref. [24]. Copyright Japanese Journal of Radiology 2010.

## 4. Discussion

Our review highlights that LC opacities can be seen throughout the gastrointestinal (GI) tract and even in the upper airways. In general X-ray, ultrasound and CT can be affected by LC opacities, but also other image modalities such as dual-emission X-ray absorptiometry (DEXA) scanning [29,30,42] and transesophageal echocardiography [28]. The heterogeneous differential diagnoses can lead to many misinterpretations (Table 4).

We did not find any examples of in vivo MRI studies regarding the effect of LC. However, one magnetic resonance imaging (MRI) study performed an in vitro trial with LC in plastic bottles filled with distilled water or edible agar. The study determined that ground LC tablets had no contrast enhancement effect on T1-weighted images; on T2-weighted images, it did not affect the signal intensity of the solvent. However, unground LC tablets may be visualized as filling defects on MRIs [43].

### 4.1. Fosrenol Administration

Some case reports described the cause of the LC opacities as whole tablets contributing to hoarseness, dysphagia, and fecaloma [4,20,22].

Furthermore, a differential diagnosis of foreign bodies results in suspicion of perforation, obstruction, intussusception, fistula formation, abdominal abscess formation, and death, and therefore, is a medically urgent situation [44]. Patients with chronic kidney disease must be encouraged to chew or crush the LC tablets, switch to Fosrenol powder, or use an alternative phosphate binder. A few studies suggest that LC should be used with caution in diverticulitis patients, as diverticulitis flare-ups may occur if the tablets are not appropriately chewed [4,18]. In particular, 1000 mg LC tablets may cause a problem for the older population, as they are 2.2 cm in diameter [20].

Radio-opacity is not exclusively linked to LC. Other drugs such as chloral hydrate, heavy metals, iodides, phenothiazines, enteric-coated pills, and solvents could also be considered if radiographic opacities are seen in imaging [45].

### 4.2. Could LC Opacities Have a Hidden Consequence?

Other studies have described the LC deposition in the gastric mucosa during endoscopy examinations as whitish lesions [46]. Shitomi et al. named the phenomenon “gastric lanthanosis” and described the pathology as reflective bright white spots by gastroscopy and eosinophilic histiocytes [41]. Another study described a high-density layer around the entire circumference of the stomach wall. These authors reported reddish-brown deposits phagocytized by macrophages histologically, and X-ray spectroscopy confirmed the presence of lanthanum in the specimen. Iwamuro et al. argued that lanthanum deposition might cause gastric erosions, ulcers, and epigastric discomfort [47]. However, a recent review described the progression and outcome of gastric lanthanosis as unknown [46]. the authors of this review propose an investigation of the clinical importance of gastric lanthanosis as a potential future research field.

The strength of this study lies in the fact that, to our knowledge, this is the first systematic review regarding LC opacities in image studies of chronic kidney disease patients. Our review gives a good overview of the consequences of differential diagnosis of LC opacities.

Nevertheless, our study also has weaknesses. Our review consists almost exclusively of case studies. Due to this feature, we were unable to determine the extent of the problem with LC opacities among patients taking LC. However, producing a visual overview of LC opacities does not require studies with a specific design. Future research might focus on the frequency of LC opacities and the most common locations, as well as the long-term effects of gastric lanthanosis.

## 5. Conclusions

Our review shows that LC opacities can vary in appearance and have been observed throughout the digestive tract of patients with chronic kidney disease. Unfortunately, the misinterpretation of these radiological opacities results in many different differential diagnoses. Therefore, we recommend that patients treated with LC should switch to an alternative phosphate binder several days prior to image testing, to avoid misinterpretation. In addition, radiologists and clinicians ordering imaging should be familiar with the effect of LC treatment on imaging, to avoid unnecessary diagnostic tests. Other tasks include making sure patients receiving LC are able to chew the tablets and, alternatively, considering a change to LC powder or an alternative phosphate binder. Further studies regarding the long-term effects of gastric lanthanosis could yield important insights.

## Figures and Tables

**Figure 1 diagnostics-12-00464-f001:**
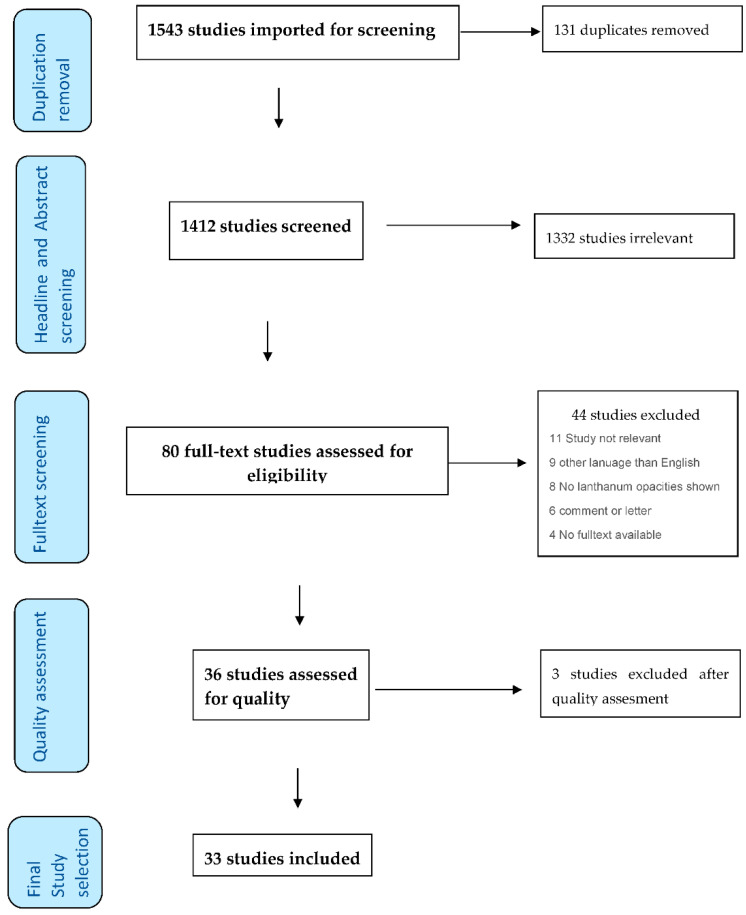
Flowchart on the included studies and the screening process according to PRISMA.

**Figure 2 diagnostics-12-00464-f002:**
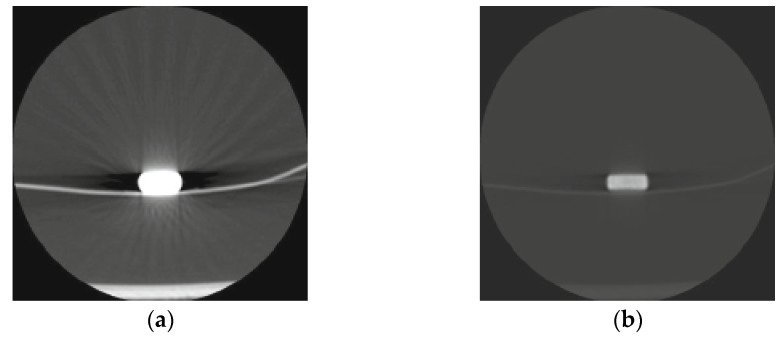
(**a**) Visual overview of lanthanum carbonate opacities. Axial CT images of lanthanum carbonate hydrate (Fosrenol chewable tablet) (window level 1500, window width −500) are shown, with a strong artifact due to a beam-hardening effect on the left, and on the right, with a window level of 15,000 and window width of 4000. (**b**) Beam-hardening artifacts on computed tomography images are also presented, caused by lanthanum carbonate hydrate in a patient on dialysis. Reprinted with permission from ref. [24]. Copyright 2010. Japanese Journal of Radiology.

**Figure 3 diagnostics-12-00464-f003:**
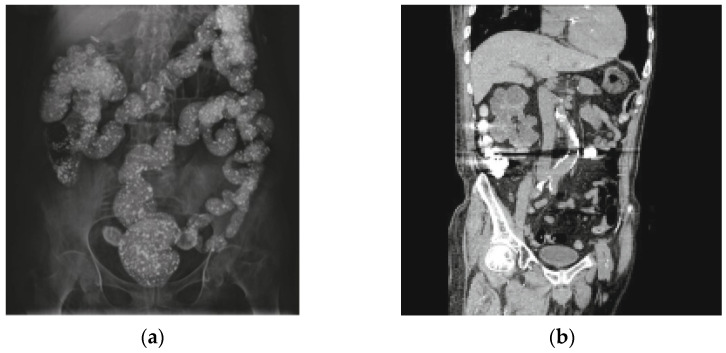
Plain abdominal radiography in which lanthanum carbonate opacities are observed in the totality of the colonic frame (**a**). (**b**) a coronal multiplanar reconstruction computed tomography (CT) image with strong artifacts caused by tablets in the ascending and transverse colon is observed. Lanthanum carbonate has a radiopaque appearance on the plain abdominal radiography. Reprinted with permission from ref. [27]. Copyright 2016 Revista Española Enfermedades.

**Figure 4 diagnostics-12-00464-f004:**
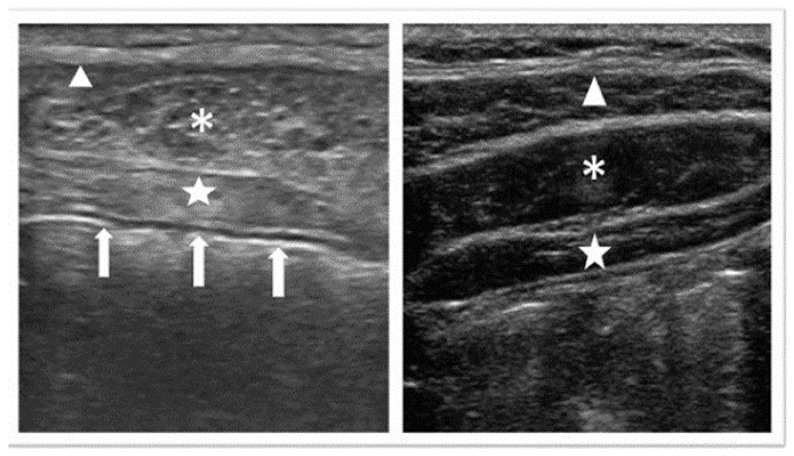
A comparison of an abdominal ultrasonogram from a patient treated with lanthanum carbonate and that of a healthy man in his 30s is presented. The portion observed as a white line below the transversus abdominis (white arrows) is the wall of the gastrointestinal tract. ∆: External oblique muscle; *: Internal oblique muscle; ☆: Transversus abdominis muscle; Lanthanum carbonate hydrate causes artifacts on ultrasound reprinted with permission from ref [28] under the Creative Commons Attribution 4.0 International License. Copyright Journal of Anesthesia 2015.

**Table 1 diagnostics-12-00464-t001:** Study and patient characteristics of the included studies.

Author	Year	Country	Studie Design	Number of Included Patients	Age	Sex	Dialyses Modality	LC Dosis/Day	LC Dispensation Form	Duration of LC Treatment
Cerny et al.	2006	Germany	Case report	1	82	M	HD	4500 mg	NR	6 mo
David et al.	2007	Germany	Case report	1	46	F	HD	2250 mg	NR	NR
C-L Chuang et al.	2007	Taiwan	Case report	1	30	F	PD	750 mg	NR	3 mo
Singanamala et al.	2008	USA	Case report	1	75	F	(acute) HD	NR	Tablets	NR (added during hospital stay)
Pafcugová et al.	2008	Czech Republic	Case report	1	77	M	HD	NR	Tablets	NR
Wu et al.	2008	Taiwan	Case report	1	56	F	PD	2250 mg	Tablets	6 mo
Badre et al.	2008	USA	Case report	1	54	M	HD	3000 mg	Tablets	3 mo
Kato et al.	2009	Japan	Case series	9	41–60 (53 ± 2)	7 M, 2 F	HD	NR	Tablets	5–112 days
Muller et al.	2009	France	Case report	1	75	F	HD	1500 mg	Tablets	NR
Connor et al.	2009	UK	Case report	1	50	M	HD	NR	NR	NR
Turkmen et al.	2009	Turkey	Case report	1	56	F	HD	3000 mg	NR	NR
Huijisoon et al.	2009	Netherlands	Case report	1	25	M	HD	2250 mg	Tablets	NR
Hayashi et al.	2010	Japan	Case report	1	68	M	HD	750 mg increased to 1500 mg after 4 days	Tablets	8 days
Chacko et al.	2010	Australia	Case report	1	66	F	PD	2250 mg	NR	NR
Hofmann et al.	2010	Germany	Case report	1	66	F	PD	NR	NR	NR
Schmitt et al.	2010	Germany	Case report	1	54	F	HD	NR	NR	NR
Fürstenberg et al.	2010	UK	Cohort study	169 → 24 prescribed LC (the rest other phosphate binders)	mean age 57 ± 16 (35.5 ± 3.6 in LC group)	46% M (37.5% in LC group)	125 PD and 44 HD (PD 66.7% in LC group)	Mean 2250 mg	NR	10.0 ± 1.3 mo
Crush et al.	2010	Ireland	Case report	1	62	M	HD	3000 mg	NR	12 mo
Walsh et al.	2011	USA	Case report	1	64	M	NR	3000 mg	NR	NR
Vila-Navarro et al.	2012	Spain	Case report	1	73	M	HD	NR	NR	NR
Muller et al.	2012	France	Case report	1	77	NR	PD	3000 mg	Tablets	NR
Kuroiwa et al.	2015	Japan	Case report	1	66	M	HD	NR	NR	NR
Salerno et al.	2015	Belgium	Case report	1	65	M	HD	1000 mg	NR	A couple of days
Ruiz-Pardo et al.	2016	Spain	Case report	1	84	F	HD	1500 mg	Tablets	1 mo
Sahbudin et al.	2016	UK	Case report	1	66	M	NR	NR	NR	NR
Nishikawa et al.	2017	Japan	Case report	1	64	F	PD	750–1500 mg	NR	6 y
Harris et al.	2017	UK	Case report	1	84	M	HD	1000 mg	NR	NR
Paranji et al.	2017	USA	Case report	1	75	M	NR	NR	Tablets	NR
Shitomi et al.	2017	Japan	Cohort study	23 (14 had CT examinations)	34–82	16 M, 7 F	3 PD and 20 HD	750 mg for all, but some had the dose reduced to 500 mg for a period.	NR	3–67 mo
Chang et al.	2018	Taiwan	Case report	1	82	F	HD	1000 mg	NR	5 mo
Fischer et al.	2019	Israel	Case report	1	69	M	NR	NR	Tablets	NR (added during hospital stay)
Shiratori et al.	2019	Japan	Case report	1	74	M	HD	NR	NR	62 mo
Galo et al.	2019	USA	Case report	1	21	M	Acute onset of chronic renal failure. Not started on dialysis at the time of the imaging study.	3000 mg	Tablets	NR

**Table 2 diagnostics-12-00464-t002:** Overview quality assessment on all studies.

	Yes	No	Unclear
Does the case/the case series mirror the whole experience of the investigator?	3	0	33
Was the exposure adequately ascertained?	20	16	0
Was the outcome adequately ascertained	35	0	1
Were alternative causes ruled out?	14	12	10
Was the case/case series described with sufficient details?	33	0	3

**Table 3 diagnostics-12-00464-t003:** practical overview including radiological study, finding, location, take-home message, and link to the article.

Author	Year	Image Study Modality (CT, X-ray, etc.)	Characteristic of Opacity	Location of Opacities	Differential Diagnosis	Take-Home Message	Link to the Article for Further Reference
Cerny et al.	2006	X-ray	Multiple, scattered radiopaque densities	Colon	Contrast medium (ruled out)	NR	Radiographic Appearance of Lanthanum|NEJM
David et al.	2007	X-ray + CT	Multiple, scattered radiopaque densities.	Intestines	NR	Lanthanum should temporarily be discontinued prior to image studies. Suggest that radiologic examinations could be used to monitor compliance in patients taking LC.	Heavy metal—rely on gut feelings: novel diagnostic approach to test drug compliance in patients with lanthanum intake|Nephrology Dialysis Transplantation|Oxford Academic (oup.com)
C-L Chuang et al.	2007	X-ray	Multiple, scattered radiopaque densities.	Intestines	Contrast medium (ruled out)	The phenomenon should be brought to the radiologist’s attention.	The Case∣A peritoneal dialysis patient with an unusual abdominal Film—ScienceDirect
Singanamala et al.	2008	X-ray	Single, round foreign body.	Aerodigestive tract	Foreign object	NR	Images in Dialysis Series Editors: Ursula C. Brewster and Mark A. Perazella: An Unexpected Finding on Chest Roentgenogram Following Hemodialysis Catheter Placement—Singanamala—2008—Seminars in Dialysis—Wiley Online Library
Pafcugová et al.	2008	X-ray + CT	Hyperdense elements with sharp edges + beam hardening.	Stomach	NR	Switch patients from lanthanum carbonate to a different phosphate binder prior to radiological examinations.	Radio-opaque appearance of lanthanum carbonate in a patient with chronic renal failure|Nephrology Dialysis Transplantation|Oxford Academic (oup.com)
Wu et al.	2008	X-ray	Multiple, scattered radiopaque densities.	Intestines	NR	Educating physicians can prevent misinterpretation.	A 56-year-old woman with starry radiopacities|Annals of Saudi Medicine (annsaudimed.net)
Badre et al.	2008	X-ray	Multiple, scattered radiopaque densities.	Upper left quadrant of abdomen→stomach.	NR	Tablets should be chewed completely prior to swallowing.	Unusual Abdominal Radio-opaque Densities in an ESRD Patient—Badre—2014—Seminars in Dialysis—Wiley Online Library
Kato et al.	2009	CT	Multiple, scattered radiopaque densities in intestines, and one case of a whole tablet seen in the terminal ileum together with digested residues in colon diverticulum.	Intestines	NR	LC should be used with caution in patients with diverticular flare-ups + in patients who are unable to chew tablets.	Accumulation of lanthanum carbonate in the digestive tracts|SpringerLink
Muller et al.	2009	X-ray + CT	Multiple, scattered radiopaque densities.	Intestines, especially in the rectosigmoid region.	NR	LC should be used with caution in patients with diverticular flare-ups + in patients who are unable to chew tablets.	confusional state associated with use of lanthanum carbonate in a dialysis patient: a case report|Nephrology Dialysis Transplantation | Oxford Academic (oup.com)
Connor et al.	2009	X-ray	Multiple, scattered radiopaque densities.	Colon	NR	Clinicians of all backgrounds should be aware of the distinctive radiological appearances seen in patients taking lanthanum carbonate. When possible, alternative phosphate binders should be temporarily employed before radiological examinations.	An unusual abdominal radiograph Postgraduate Medical Journal (bmj.com)
Turkmen et al.	2009	X-ray	Multiple, scattered radiopaque densities.	Throughout the ascending and transverse colon segments	NR	NR	An unusual hurdle to renal transplantation: speckled abdominal opacities induced by lanthanum carbonate—Turkmen—2010—Internal Medicine Journal—Wiley Online Library
Huijisoon et al.	2009	X-ray	Multiple, scattered radiopaque densities.	Intestines	Contrast medium (ruled out)	Patients should chew the tablets sufficiently.	Dustri Online Services
Hayashi et al.	2010	CT	Multiple, hyperdense elements with sharp edges + beam hardening.	Intestines	Foreign bodies	Radiologists should be familiar with the appearance of LC. Temporarily switch to a different phosphate binder before the radiological examination. If CT or X-ray is necessary, perform before administration of LC.	Beam-hardening artifacts on computed tomography images caused by lanthanum carbonate hydrate in a patient on dialysis SpringerLink
Chacko et al.	2010	X-ray	Multiple, scattered radiopaque densities.	Intestines	Sclerosing peritonitis, tuberculosis, and lead ingestion	Awareness of LCs radiopaque features will prevent unnecessary investigations.	Christmas lights in the gastrointestinal tract The Medical Journal of Australia (mja.com.au)
Hofmann et al.	2010	X-ray	Multiple, scattered radiopaque densities.	Colon	Oral or rectal contrast medium	NR	Colonic Opacification in a Patient with End-Stage Kidney Disease—Gastroenterology (gastrojournal.org)
Schmitt et al.	2010	X-ray + CT	Multiple, scattered radiopaque densities.	Intestines	NR	Proposes that Lanthanum should temporarily be discontinued prior to image studies	Layout 1 (nih.gov)
Fürstenberg et al.	2010	DEXA	Multiple, scattered radiopaque densities.	Intestines	NR	LC opacities in the GI tract can lead to erroneous overestimation of bone mineral content.	Overestimation of Lumbar Spine Calcium with Dual Energy X-Ray Absorptiometry Scanning due to the Prescription of Lanthanum Carbonate in Patients with Chronic Kidney Disease—Abstract—American Journal of Nephrology 2010, Vol. 32, No. 5—Karger Publishers
Crush et al.	2010	X-ray	Multiple, scattered radiopaque densities.	Colon	Contrast medium (ruled out)	Physicians should be informed about the phenomenon.	Perplexing plain abdominal X-ray Gut (bmj.com)
Walsh et al.	2011	DEXA	Multiple, scattered radiopaque densities.	Colon	Contrast medium	Lanthanum in the colon falsely increases BMD measurement. Suggests temporarily switching to a different phosphate binder before DXA scan if possible.	ClinicalKey
Vila-Navarro et al.	2012	X-ray	Multiple, scattered radiopaque densities.	Colon and appendix	Contrast medium (ruled out)	NR	07_IPD_2384-Vila.ing_Maquetación 1 (grupoaran.com)
Muller et al.	2012	X-ray + CT	Radiopaque puddle with dilation and fecal impaction.	Stomach	NR	Chewing the tablet is essential. Caution is required if the patient suffers from constipation as he is at risk of developing a fecaloma or encephalopathy if the lanthanum serum content increases and crosses the brain blood barrier.	Radio-opaque fecal impaction and pseudo-occlusion in a dialyzed patient taking lanthanum carbonate—Muller—2012—Hemodialysis International—Wiley Online Library
Kuroiwa et al.	2015	UL + CT	Hyperecchoic signal in the intestines.	Intestines	NR	Anesthesiologists who perform transesophageal echocardiography and the abdominal US need to be familiar with the characteristics of LCH.	Lanthanum carbonate hydrate causes artifacts on ultrasound (nih.gov)
Salerno et al.	2015	CT	Multiple, hyperdense elements with sharp edges + beam hardening.	Intestines	Foreign bodies, intestinal bleeding.	One should pay more attention to the patient’s food habits as well as the drugs he is receiving to avoid misinterpretations of the radiological imaging studies.	The Risk of Mistaking Intestinal Lanthanum Carbonate for Intestinal Bleeding on CT (nih.gov)
Ruiz-Pardo et al.	2016	X-ray	Multiple scattered radiopaque densities.	Intestines	Contrast medium	NR	10_IPD_3822_Ruiz.Ing.indd (reed.es)
Sahbudin et al.	2016	DEXA + X-ray	Multiple scattered radiopaque densities.	Intestines	NR	Alternative phosphate binders should be used.	Lanthanum carbonate in chronic renal failure|The BMJ
Nishikawa et al.	2017	CT	High-density layer around the entire circumference of the stomach wall.	Stomach wall	NR	Clinicians should be aware of this clinical condition as a possible cause of nausea	Lanthanum deposition in the gastric mucosa in a patient undergoing hemodialysis|QJM: An International Journal of Medicine|Oxford Academic (oup.com)
Harris et al.	2017	X-ray + CT	Multiple, scattered radiopaque densities (X-ray) + attenuation in the bowel wall and scattered opacities in the stomach and intestines (CT).	Intestines	Plebosclerotic colitis, hemorrhage	In Plebosclerotic colitis, it is unusual for the whole colon to be involved.	A case of lanthanum carbonate ingestion thought to be Plebosclerotic colitis on CT imaging and abdominal radiograph—ScienceDirect
Paranji et al.	2017	X-ray	A radiopaque coin-shaped foreign body.	aerodigestive tract	Foreign object	Tablets should be crushed/chewed sufficiently before swallowing due to the risk of aspiration.	All that glitters is not gold: A case of lanthanum carbonate aspiration—Suchitra Paranji, Neethi Paranji, Adam S Weltz, 2017 (sagepub.com)
Shitomi et al.	2017	CT	High-attenuating lines in the stomach wall.	Stomach	Non—proven to be lanthanum with biopsies	Coined lanthanum carbonate accumulation in the gastric mucosa “gastric lanthanosis”. The clinical implication is unknown.	Gastric lanthanosis (lanthanum deposition) in dialysis patients treated with lanthanum carbonate—Shitomi—2017—Pathology International—Wiley Online Library
Chang et al.	2018	X-ray + CT	Multiple, scattered radiopaque densities.	Colon	Barium contrast	Clinicians may take its radiopaque characteristic as an advantage to assess the patient’s drug adherence.	Starry-sky bowels|SpringerLink
Fischer et al.	2019	X-ray + CT	Multiple, scattered radiopaque densities.	Stomach, small intestines, and rectum.	Foreign bodies	Familiarity with the unique appearance of lanthanum carbonate on imaging can prevent misinterpretation of imaging modalities.	Page loading—ClinicalKey
Shiratori et al.	2019	CT	Hyperdensity areas in the stomach wall.	Stomach	NR	Esophagogastroduodenoscopy was performed showing white regional lesions due to Lanthanum carbonate deposition.	Lanthanum deposition in the gastric mucosa in a patient treated with hemodialysis BMJ Case Reports
Galo et al.	2019	UL + X-ray + CT	Multiple, scattered radiopaque densities.	Intestines	Foreign bodies and other drugs that typically cause radiopaque appearance.	The chewable tablets should be fully chewed or crushed prior to ingestion. One should first rule out abdominal diseases before prescribing lanthanum carbonate. Recognition of lanthanum carbonate artifacts can avoid extensive and costly use of medical resources for this benign condition.	Lanthanum-Induced Radiopaque Intestinal Precipitates: A Potential Cause of Intestinal Foreign Bodies (nih.gov)

**Table 4 diagnostics-12-00464-t004:** Differential diagnosis of lanthanum carbonate opacities.

Differential Diagnosis
Small metal objects such as small coins [20,35]
Contrast [25,27,30,34,35,36,37,38,39]
Sclerosing peritonitis [10]
Tuberculosis [10]
Lead ingestion [10]
Intestinal bleeding [11,12]
Phlebosclerotic colitis [11]

## Data Availability

Links to the different studies included in the review are listed in Table 3.

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
