# Peer review of "Lanthanum Carbonate Opacities—A Systematic Review"

_diagnostics, 2022, doi:10.3390/diagnostics12020464_

Round 1
Reviewer 1 Report
No further request from my side
Author Response
Dear reviewer,
thank you very much for your kind words.
Kind regards
Jan Kampmann
Reviewer 2 Report
The authors provided a review study regarding Lanthanum Carbonate Opacities detected during radiological investigation. The current manuscript is of great importance because they allow radiologist and clinicians to understand the causes of the opacities. In addition to that, the manuscript is well organized and provides valuable information. However, some minor corrections need to be made; some clarification is also necessary before the acceptance.
Abstract
1) The study background and problem are not clearly stated
2) Data collection method, criteria, and time frame are not included
3) The conclusion is a repetition of known information
Introduction
1) The introduction needed further improvement. This topic is multidisciplinary. Therefore, the introduction should cover background regarding Lanthanum Carbonate density and other characteristics
2) Clinical indication and usage and alternatives should be included
3) Provided further data regarding the absorption LC
4) On Page 2, line 47, different imaging modalities I not accurate. The study provided transmission imaging modalities. Please check
Materials and Methods
1) Provide details regarding data analysis
Results
1) It is helpful to present the previous studies in the Tables on a chronological basis
Discussion
1) The discussion requires further improvement as a systemic review.
2) Provide details regarding the clinical implication of LC opacities and how they can affect or obscure findings.
Conclusions
1) Write a conclusion based on study findings, not just a repetition of previous studies
2) Provide the added value of this study
3) Suggestion for clinicians and researchers

Author Response
Please see the attachment

This manuscript is a resubmission of an earlier submission. The following is a list of the peer review reports and author responses from that submission.
Round 1
Reviewer 1 Report
Please see attached.

Reviewer 2 Report
The authors conducted a literature review regarding lanthanum carbonate image findings. Overall, the major problem for this review is that this topic is rather obsolete and not worthy of investigation. As the authors stated in the introduction, several others already addressed similar issue in the literature, and this work essentially adds little to the current knowledge. Other concerns are listed below.
- The utilized quality assessment method typically applies to original investigations only, but not for case series or reports. Such usage can be inappropriate.
- The images reproduced in this manuscript seem not obtaining prior consent for re-use, which can be an important problem related to publication ethics.